# Two-step offer and return of multiple types of additional genomic findings to families after ultrarapid trio genomic testing in the acute care setting: a study protocol

Sophie E Bouffler ,[1] Ling Lee,[2,3] Fiona Lynch,[3,4] Melissa Martyn,[2,3] Elly Lynch,[2,5] Ivan Macciocca,[4,5] Lisette Curnow,[5] Giulia McCorkell,[1,4] Sebastian Lunke,[4,5] Belinda Chong,[5] Justine E Marum,[5] Martin Delatycki,[4,5] Lilian Downie,[3,5] Ilias Goranitis,[1,6] Danya F Vears,[3,4] Stephanie Best,[1,7] Marc Clausen,[8] Yvonne Bombard ,[8,9] Zornitza Stark,[1,5] Clara L Gaff[2,3]

**Correspondence to**
Dr Clara L Gaff;
clara.gaff@melbournegenomics.org.au

## ABSTRACT

**Introduction** As routine genomic testing expands, so too does the opportunity to look for additional health information unrelated to the original reason for testing, termed additional findings (AF). Analysis for many different types of AF may be available, particularly to families undergoing trio genomic testing. The optimal model for service delivery remains to be determined, especially when the original test occurs in the acute care setting.

**Methods and analysis** Families enrolled in a national study providing ultrarapid genomic testing to critically ill children will be offered analysis for three types of AF on their stored genomic data: paediatric-onset conditions in the child, adult-onset conditions in each parent and reproductive carrier screening for the parents as a couple. The offer will be made 3–6 months after diagnostic testing. Parents will have access to a modified version of the Genetics Adviser web-based decision support tool before attending a genetic counselling appointment to discuss consent for AF. Parental experiences will be evaluated using qualitative and quantitative methods on data collected through surveys, appointment recordings and interviews at multiple time points. Evaluation will focus on parental preferences, uptake, decision support use and understanding of AF. Genetic health professionals' perspectives on acceptability and feasibility of AF will also be captured through surveys and interviews.

**Ethics and dissemination** This project received ethics approval from the Melbourne Health Human Research Ethics Committee as part of the Australian Genomics Health Alliance protocol: HREC/16/MH/251. Findings will be disseminated through peer-review journal articles and at conferences nationally and internationally.

## INTRODUCTION

Additional findings (AF) refer to results from genomic testing that may have medical value and/or utility and are found in a deliberate search unrelated to the original reason

### STRENGTHS AND LIMITATIONS OF THIS STUDY

⇒ This is a prospective national pilot study across 17 clinical sites exploring the offer and receipt of three types of additional findings (AF) following genomic testing in the acute care setting.

⇒ This study will investigate the practicability, acceptability and value of offering AF analysis to parents and children using stored genomic data 3–6 months after diagnostic testing.

⇒ Results from this study can inform future health system implementation of analysis for AF.

⇒ A process evaluation will be undertaken to investigate parental decision-making and preferences for health service delivery.

⇒ This study explores AF specifically in the context of acute neonatal and paediatric care.

for testing.[1 2] Such findings may include adult-onset and paediatric-onset treatable and untreatable disorders, reproductive carrier status and pharmacogenomic testing. Research participants consistently express a high level of interest in receiving AF.[3–5]

The American College of Medical Genetics and Genomics (ACMG) initially advocated the return of adult-onset AF as a mandatory part of genomic testing regardless of the patient's age,[1] with updated guidelines suggesting an opt-in approach with a minimum list of 73 genes to be reported.[6] The offer of paediatric-onset AF for children undergoing genomic sequencing has been explored, often as a prelude to developing newborn screening programmes.[7 8] Stored genomic data can also be used for reproductive genetic carrier screening. As carrier

screening becomes more common,[9] it is a natural extension for parents undergoing trio genomic testing to be offered this additional information as they are typically of childbearing age. A recent systematic review[10] summarised the position of several countries, with many international bodies, including the Canadian College of Medical Geneticists and the European Society of Human Genetics (ESHG), supporting a more conservative approach and recommended that analysis and reporting from genomic sequencing should be focused on the primary reason for testing.[11 12] In Australia, analysis is largely restricted to genes known to be associated with the patient's primary clinical indication for testing.

Determining whether and how to deliver AF within health services continues to be debated by professional bodies. International studies have described models whereby the offer of AF is made at the time diagnostic testing is initiated.[13–15] There are concerns that this model adds to the complexity of pre-test counselling at a time when families are already overwhelmed by having to make many complex decisions. This is particularly pertinent in the acute neonatal and paediatric care setting, when parents are dealing with decision-making for their critically ill child in a highly stressful, time pressured environment. Concerns have been raised about parents' ability to process and retain information and the impact of genomic results on family functioning in this setting.[16–21] Emerging evidence supports the validity of these concerns,[22] and in one study the offer of AF deterred families from receiving genetic testing when a diagnostic result would be in the child's best interest.[23] The ESHG has more recently acknowledged the method proposed by the French Society of Predictive and Personalised Medicine, which recommends a multistep approach to consent for AF,[24] but note the need for further empirical evidence.[25]

The Melbourne Genomics Health Alliance described a pilot study trialling a two-step offer for AF analysis on stored data in adult patients, with the offer occurring subsequent to the primary result return.[26] This model allows for temporal separation of the offer and avoids burdening patients with complex decisions at a time of high stress, while still offering them the opportunity to receive more health information from their stored genomic data.

The routine provision of AF in a two-step model could involve a significant investment of time and resources.[27] As evidence reinforcing the importance of providing detailed information during the informed consent process in this space emerges[28] it is essential to explore tools that will support this. Digital decision support tools have the potential to alleviate some of the resource constraints when implemented in a genomic care pathway, alongside in-person encounters.[29 30]

As genomic sequencing is increasingly incorporated into different clinical settings there continues to be a need for evidence to inform decisions about how healthcare systems should manage AF. Storage of genomic data enables analysis for AF as a separate decision after diagnostic testing is complete.

The Acute Care Genomics (ACG) study[31] is evaluating ultrarapid trio whole genome sequencing for critically ill infants and children on a national scale. The study cohort includes parents and children who have stored whole genome sequencing data from trio testing. This cohort provides the opportunity to understand the service implications of offering and returning multiple AF in acute care on a national scale, as well as parental preferences for receiving this information, across multiple genetic services.

### Study aim

We describe a prospective national study offering analysis for multiple types of AF on stored data to children and their parents undergoing trio genomic testing via a two-step delivery model. This study uses process evaluation to examine decision-making, decision support, counselling processes, as well as resource and service implications.

## METHODS AND ANALYSIS
### Eligibility and recruitment

This study will recruit families via a two-step model as described in a previous study.[26] The ACG study will offer up to 250 families ultrarapid genomic trio/duo testing at 17 clinical sites around Australia between July 2020 and April 2022. As part of their participation in the ACG study, parents will be approached 3–6 months after disclosure of the child's ultrarapid genomic testing result to be offered analysis for AF using their stored genomic data. Research has shown that parents commonly try to conceive shortly after experiencing a perinatal loss.[32 33] To best support families in this situation, parents who enquire seeking recurrence or reproductive risk information before the study-prompted offer of AF may have the opportunity for earlier analysis. This study will run from July 2020 to June 2023. Study data will be managed using REDCap electronic data capture tools hosted at the Murdoch Children's Research Institute, Melbourne, Australia.[34 35]

Families will be excluded where the child is still a hospital inpatient, or if the proband's death is under coroner's investigation. Under exceptional circumstances, if the referring clinical team deem it inappropriate to approach a family for psychosocial reasons they will be excluded. Examples of such circumstances may be families experiencing high levels of psychological distress, cases where the parents are in the process of relinquishing care of the child, or if one parent is no longer involved in the child's care.

Families will be eligible to receive three types of AF:
1. Paediatric-onset AF in the child.
2. Adult-onset AF in each parent.
3. Expanded couple genetic carrier screening.

Where the child is deceased, paediatric-onset AF will not be offered but testing will be offered to the parents. Expanded couple genetic carrier screening will not be

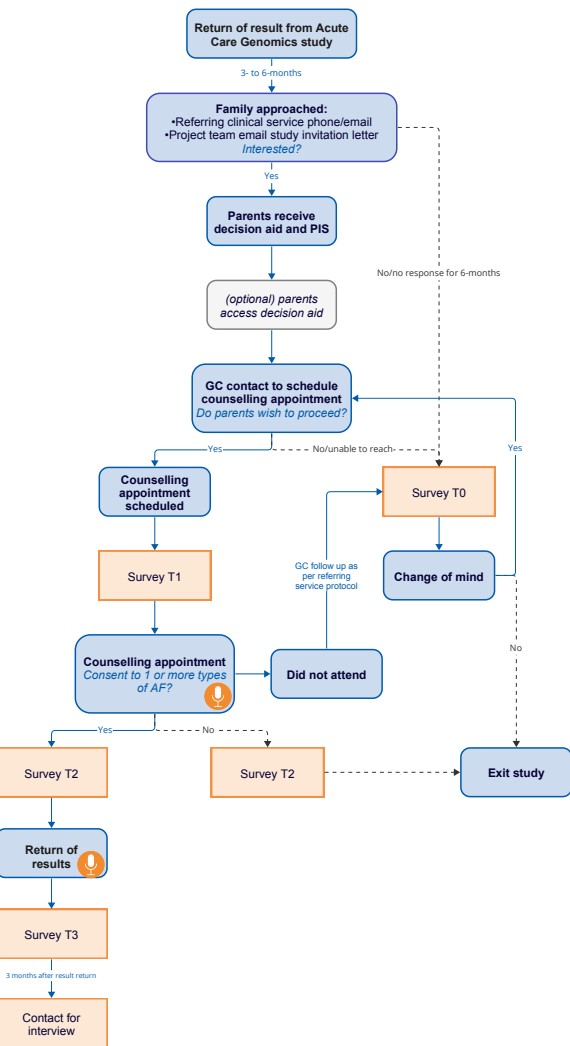

**Figure 1** Recruitment flow diagram including survey time points. Microphone denotes timing of appointment recordings. AF, additional finding; GC, genetic counsellor; PIS, patient information sheet.

offered for duos (one parent and child), or where the parents are separated.

## Study design

The study design is outlined in figure 1.

The study has been designed to minimise distress to parents wherever possible, noting the increased sensitivity around parents in the acute care setting. To reflect expected real-world service delivery and facilitate continuity of care, families will be approached by the original referring clinical service and the same provider who saw them for the initial testing where feasible. Alternatively, the study team will send an approach letter via email including a link to a survey asking parents for their expression of interest. Parents will be approached in separate emails if the referring team so advise.

Parents who express interest will receive login details for a web-based decision support tool and a copy of the patient information sheet. The counselling team will subsequently contact families to arrange a pre-test genetic

counselling appointment to discuss consent. All families must meet with a genetic health professional to proceed with receiving AF.

## Decision support

The study will deploy an interactive web-based decision support tool, called Genetics Adviser,[36] as an adjunct to a genetic counselling appointment. The tool is an expanded version of the Genomics ADvISER, (a genomics decision AiD about Incidental SEquencing Results), which has been shown to effectively augment genetic counselling and improve patient knowledge and decisional confidence.[30 37] The Genetics Adviser is a comprehensive, interactive, patient-centred platform aimed at providing lay language digital content to support informed decision-making, offer pre-genetic testing and post-genetic testing support, and facilitate return of results. The tool can be tailored for particular use cases as well as a range of clinical and population testing scenarios.[36] For this protocol, the Genetics Adviser was customised to focus on pre-test counselling and education for AF with the support of genetic counselling in this process.[38] Content has been adapted in collaboration with the Genetics Adviser team to enable the delivery of tailored information on the three types of AF offered in this study with a focus on using plain language for the content and videos. To minimise distress to parents of children who have died, an alternative version of the tool will be used for these parents, which omits the section on paediatric-onset AF.

Each parent will receive separate login details for the Genetics Adviser and will be encouraged to work through the tool individually prior to the pre-test genetic counselling appointment.

## Genetic counselling and consent

All families will receive genetic counselling. Pre-test and post-test genetic counselling appointments will be conducted in person or via telehealth. Families will be asked for consent to audio record pre-test and post-test genetic counselling appointments following an approved verbal consent script.

Genetic health professionals who will conduct pre-test counselling for AF will be invited to an online education workshop facilitated by genetic counsellors with experience delivering AF in a research setting. The workshop will include lessons from previous studies offering AF, small group role plays of AF counselling scenarios, and detailed study logistics. There will be a pre-workshop self-directed module including two video recorded simulation pre-test counselling appointments using trained counsellors and actors. The participating genetic health professionals will be provided with talking points and tips for counselling families to ensure that the information given and language used is consistent between services, as well as a visual aide summarising the different types of AF (figure 2).

All participants will provide informed consent for data collection as part of their involvement in the ACG study.[31]

| | Types of Information Offered | | |
|---|---|---|---|
| | **Childhood-onset Additional Findings** | **Adult-onset Additional Findings** | **Genetic Carrier Screening** |
| **Who is the information about?** | Your child | Both parents separately | Both parents together |
| **What information is being looked for?** | Conditions that affect children. These may or may not have a known treatment or intervention to improve your child's health | Conditions that may develop in adulthood with a known treatment or intervention | Increased chance as a couple of having a child with a different severe or life-threatening genetic condition. Some are treatable, others are not |
| **Examples** | Muscular dystrophy, some types of blindness or deafness | Hereditary breast and ovarian cancer, hereditary bowel cancer, genetic heart conditions | Carrier for cystic fibrosis, sickle cell disease, some forms of developmental delay |
| **Impact** | Can reveal your child's chance of developing another genetic condition | You may need more tests and procedures | Can impact pregnancy planning |

**Figure 2** Visual aide for use during pre-test genetic counselling appointment detailing the different types of additional findings.

This consent encompasses the research component of the AF study, with parents providing separate clinical consent to the different types of AF.

Clinical consent for AF will be captured via the REDCap e-consent framework.[39] For the majority of families, consent will be documented using a single clinical e-consent form that displays all of the AF offerings the family are eligible for. Where the referring team indicate the decision for adult-onset AF will be made in separate appointments, the genetic health professional will have access to individual e-consent forms for each parent. Consent from one parent is required to proceed with analysis for paediatric-onset AF in the child; however, the genetic health professional will discuss this decision with both parents and ensure there is a consensus where possible as per standard clinical practice. Both parents must provide consent for couple genetic carrier screening to proceed.

All results will be returned by a genetic health professional in person or via telehealth/phone. All results will be returned simultaneously, unless the couple are pregnant and have requested carrier testing, in which case the results may be returned early to facilitate reproductive planning.

### Gene lists
All gene lists have been developed in previous studies, using structured frameworks and expert groups to define content. The gene list for the Australian Reproductive Carrier Screening Project (Mackenzie's Mission_Reproductive Carrier Screening) will be used for couple carrier screening.[40] Paediatric-onset additional findings (Additional findings_Paediatric) will encompass genes tested as part of the BabySeq project,[41] NC NEXUS study[42] and Baby Beyond Hearing project.[7] The final list includes

treatable and untreatable conditions affecting children. The gene list for adult-onset additional findings (Additional findings_Adult) is based on the list created for the Melbourne Genomics Additional Findings flagship,[26] which includes genes associated with clinically actionable conditions where a publicly funded management pathway exists. This list was updated to include genes on the ACMG V.3.0 Secondary Findings list.[6]

Gene lists to guide analysis for the different types of AF are available on PanelApp Australia (https://panelapp.agha.umccr.org/).[43 44] The content on this platform is openly available.

### Laboratory protocol
The laboratory will receive separate requests for each type of AF to which a family has given consent. The laboratory will analyse the stored genomic data according to National Association of Testing Authorities clinically accredited procedures. Variants will be classified based on ACMG classification guidelines[45] and only pathogenic or likely pathogenic variants will be reported. All variants identified for reporting will be discussed in a multidisciplinary team meeting, including clinical geneticists, other relevant medical subspecialists, genetic counsellors, bioinformaticians and molecular geneticists. Variants of uncertain significance will not be reported, in line with population screening guidelines.[46] The standard turnaround time for AF reports will be 12 weeks. Reproductive carrier screening results will be expedited if clinically indicated.

Where the child has previously been diagnosed with an autosomal recessive, X-linked or inherited dominant condition, increased reproductive risk will be included on the couple carrier screening report, regardless of whether the gene is contained in the carrier screening panel.

### Evaluation
This study will answer the following questions:
1. What is the uptake of the different types of AF?
2. What influences people's decisions to accept or decline the different types of AF?
3. How is the decision support tool used?
4. What are stakeholder perspectives of the decision support tool?
5. To what extent do counselling tools and processes support understanding of AF?
6. What are stakeholders' (patients and genetic health professionals) views of offering AF in the acute care setting?

### Clinical and genetic counselling data collection
We will collect data on how many families are offered, accept and decline AF analysis and at which stage of the process this occurs. We will collect data on the types of AF families choose to have and the outcomes of analysis. Data will also be collected on laboratory turnaround

times. Pre-test and post-test genetic counselling appointments will be audio recorded.

### Surveys: parents

Participants will be invited to prospectively complete surveys throughout the study (see figure 1). All surveys will be delivered electronically to each parent via the study's REDCap database. Participants will be invited to complete the first survey (T1) prior to the pre-test counselling appointment, the second (T2) after deciding whether to accept any or decline all AF and the third (T3) following result return. Survey responses will be linked between time points. If participants decline before counselling, they will receive a decliner survey (T0); immediately for active decliners (parents who explicitly decline AF) or 6 months after approach for passive decliners (parents who do not respond to contact from the clinical or study team).

Survey measures are outlined in table 1. T1 will capture perspectives on the decision tool, parent understanding of the AF offer and baseline psychometrics. T2 will evaluate processes and reasons for decision-making as well as parental values and understanding of analysis for AF post-counselling. T3 will capture information on the participant's values, experience and understanding of AF results.

### Parent interviews

Participants will be invited to provide their contact details during any of the surveys if they are willing to be contacted for an interview to discuss their experience of the study in greater detail. Those who do so will be contacted a minimum of 3 months after result return or after they decline AF and offered the option to participate in a semi-structured interview, either by phone or via an online platform (see figure 1). The interviews will explore participants' experiences with the web-based decision support tool, genetic counselling appointment, consent process and result disclosure. Interviews will be audio recorded, transcribed, deidentified and analysed using inductive content analysis.

### Genetic health professional evaluation

The perspectives of genetic health professionals (genetic counsellors and clinical geneticists) implementing AF in this study will be evaluated using a mixed methods study. Participating health professionals will be identified by the study team and receive a validated survey[47] prior to commencing the AF study with families and on completion of result return to the final participants. The survey aims to establish practitioners' perspectives of the implementation outcomes of acceptability, appropriateness and feasibility.[48] Follow-up semistructured interviews will be undertaken through an online platform to investigate influences on clinicians' perspectives and to identify enablers to the adoption of AF for practitioners. Transcripts will be audio recorded and transcribed before deductive and inductive data analysis is undertaken.

### Data analysis

Overall AF uptake will be reported as the number of families who accept any type of AF divided by the total offered any AF. Uptake of each type of AF will also be reported, noting that the denominator for each AF offer may differ.

Quantitative data from surveys (Likert scales, categorical items) will be analysed using standard statistical methodology. If sample sizes for each group provide sufficient power, multivariate statistical analyses will be performed to explore factors that may influence uptake (eg, received primary diagnosis or not, family completion status). Economic implications associated with the delivery of AF and the effect on follow-on care pathways will be evaluated using health economic modelling methods, subject to group size. Quantitative data on decision support tool uptake and utilisation will be analysed. Uptake will be measured by determining the number of parents who accessed the Genetics Adviser divided by the number of parents who were invited. Utilisation will be measured by the time parents took to work through the Genetics Adviser and the drop-off rate.

Qualitative data will include open-text survey responses and audio recordings (of interviews and genetic counselling sessions). Open-text survey responses will be analysed iteratively, using inductive content analysis, and used to explain the quantitative survey findings. The approach to analysis of counselling transcripts will be informed by responses to surveys and decision support tool utilisation metrics. It is anticipated that a hybrid deductive-inductive content analysis approach will be used to code counselling interactions to explore the impact of use of the decision support tool on counselling and decision-making and identify counselling challenges (specific to the acute care setting).

### Patient and public involvement

The design and delivery of this study has been heavily informed by previous consumer focused studies,[7 26] incorporating learnings from participant responses and feedback gathered in these studies. The study also uses Genetics Adviser, a decision support platform with extensive public input into design and development.[30 36–38]

## DISCUSSION

This protocol describes the offer and return of three categories of AF to families participating in a study examining national implementation of ultra-rapid genomic diagnostic testing in the acute care setting. We will use a two-step approach to offer parents analysis for paediatric-onset conditions in their child, adult-onset conditions in themselves and reproductive carrier screening as a couple 3–6 months after receiving genomic test results for their child.

The challenges of providing multiple types of AF have not been previously investigated in families undergoing testing and the acute care setting raises unique issues. Service models are needed that support the

**Table 1** Survey measures

| Measure | Description | T1 | T2 | T3 | T0 |
|---|---|---|---|---|---|
| Decision support tool user assessment | Six study-specific questions about decision support use, clarity, relevance, bias, other information needs and areas of improvement. | X | | | |
| Knowledge of AF | Seven study-specific questions to determine a participant's understanding of AF offered. | X | X | | |
| State Trait Anxiety Inventory (STAI-AD) | 26-item scale measuring state and trait anxiety.<br>Copyright 1968, 1977 by Charles D. Spielberger. All rights reserved in all media. Published by Mind Garden. *www.mindgarden.com*<br>Note: The six questions that form the STAI-6 short form[49] are repeated at each time point. If participants complete any of the remaining 20 items in any survey they will not be asked these items again. | X | X | X | X |
| Health behaviour | A shortened version of the Threatening Medical Situations Inventory.[50–52] | X | X | | X |
| Demographics | Age, gender, education, income, language, marital status, no of children, prior experience with genomic testing, family planning, private health insurance status, postcode.<br>Note: if participants complete demographic questions in any survey they will not be asked again. | X | X | X | X |
| Decision recall | One study-specific question to assess whether parents recall their decision. | | X | X | |
| Difficulty and deliberation of decision | Two study-specific questions to assess how difficult it was and how long it took parents to decide which AF to receive (T2) or to decline (T0). | | X | | X |
| Reasons for accepting or declining | Three study-specific questions for each type of AF offered addressing reasons for the participant's decision. Participants are asked to rate a selection of reasons on a 5-point Likert scale and asked to comment if there are other reasons not listed.<br>Note: separate questions for acceptors and decliners of the different types of AF | | X | | |
| Decisional conflict scale | 16-item scale measuring decisional conflict.[53] | | X | | |
| Acceptable information needs | Two study-specific questions about whether use of the decision tool alone without genetic counselling would provide enough information to make a decision. | | X | | |
| Genetic counselling satisfaction | Seven-item scale addressing patient satisfaction and service quality in the clinical genetics setting.[54] | | X | X | |
| Willingness to pay | Dynamic triple-bounded dichotomous choice contingent valuation, also known as a 'bidding game' to assess the value participants place on the information that comes from analysis for AF. | | X | X | |
| Results recall and understanding of results | Four study-specific questions to determine if participants recalled their results correctly, and if they understand what these results mean. | | | X | |
| Future planning | Five study-specific questions on future planning for themselves, their child and their family based on the AF results received. | | | X | |
| Decision regret scale | Five-item scale to measure regret after a healthcare decision.[55] | | | X | |
| FACToR questionnaire | 12-item questionnaire adapted to assess the psychosocial impact of returning genomic test results.[56] | | | X | |
| Genomics outcome scale | Six-item scale based on the Genetic Counselling Outcome Scale adapted to assess outcomes of genetic counselling and patient empowerment.[57] | | | X | |
| Service delivery preferences | Three study-specific questions addressing how and when the offer for AF should be made. | | | X | |
| Value of offering AF | One study-specific question on the value of information that comes from analysis for AF. Participants are asked to rate different factors on a 4-point Likert scale from extremely valuable to not valuable. | | | X | |
| Reasons for declining | One study-specific question exploring the reasons for declining. Participants are presented with 13 reasons and asked to rate how much each one influenced their decision on a scale of 1 (did not influence) to 5 (strongly influenced). | | | | X |

AF, additional findings; FACToR, Feelings About genomiC Testing Results; STAI-6, Six-item State-Trait Anxiety Inventory; STAI-AD, State-Trait Anxiety Inventory for Adults; T0, passive or active decliner survey; T1, pre-test counselling survey; T2, post-decision for AF survey; T3, post-result return survey.

family's autonomy and informed decision-making and are feasible to deliver within a constrained health system. Some key decision points in designing this service included the timing of the offer in relation to the trajectory of the child's illness; design of online ordering and consent forms that account for different family structures and potential privacy concerns; and whether to offer paediatric or adult-onset AF to adolescents and how to support involvement in decision-making in this particular group.

The sample size for this protocol is predetermined by recruitment into the larger ACG study. This will be considered when reporting the findings of our study and we acknowledge that there may still be a need for larger, purposefully designed studies in the future.

Evidence is currently lacking to guide policy and decisions on service delivery models for AF, including uptake; patient experience and preferences; and resource requirements. We will perform a multidisciplinary evaluation to generate comprehensive data across multiple clinical sites, nationally, which will inform future service delivery.

## Ethics and dissemination

This project has received ethics approval from the Melbourne Health Human Research Ethics Committee as part of the Australian Genomics Health Alliance protocol: HREC/16/MH/251. All participants will provide written informed consent for data collection, and if relevant, to receive analysis for AFs. Findings from this study will be published in peer-reviewed journals and presented at national and international conferences. All participants have provided informed consent to be involved in this study.

### Author affiliations
[1]Australian Genomics Health Alliance, Parkville, Victoria, Australia
[2]Melbourne Genomics Health Alliance, Walter and Eliza Hall Institute of Medical Research, Melbourne, Victoria, Australia
[3]Murdoch Children's Research Institute, Parkville, Victoria, Australia
[4]Department of Paediatrics, Faculty of Medicine, Dentistry and Health Sciences, The University of Melbourne, Melbourne, Victoria, Australia
[5]Victorian Clinical Genetics Services, Murdoch Children's Research Institute, Parkville, Victoria, Australia
[6]Health Economics Unit, Centre for Health Policy, Melbourne Schoold of Population and Global Health, The University of Melbourne, Melbourne, Victoria, Australia
[7]Department of Health Services Research, Peter MacCallum Cancer Centre, Melbourne, Victoria, Australia
[8]Genomics Health Services Research Program, St. Michael's Hospital, Unity Health Toronto, Toronto, Ontario, Canada
[9]Institute of Health Policy, Management and Evaluation, University of Toronto, Toronto, Ontario, Canada

**Contributors** Study concept and design was conceived by ZS, CLG, SL. SEB, LL, MM, EL, LD, IG, DFV, SB, BC, JEM, MD, MC and YB contributed to specific evaluation parts of the study and refining study design. IM, LC, GM, EL, LL and SEB were involved in design and delivery of the training for genetic counsellors. LL, FL, DFV, IG, SB and MM will evaluate the data. SEB prepared the first draft of the manuscript. All authors contributed to the drafting of the manuscript, read and approved the final manuscript. In addition to the authors listed, the broader Acute Care Genomics program scientific, clinical and diagnostic membership had input into this protocol.

**Funding** This work was supported by the Australian Government's Medical Research Future Fund grant number GHFM76747 and the Royal Children's Hospital Foundation grant number 2020-1259. The authors are funded by Australian Genomics, the Melbourne Genomics Health Alliance (grant number N/A), the State Government of Victoria (Department of Health and Human Services; grant number N/A) and the Australian Government's Medical Research Future Fund. Australian Genomics is funded by the National Health and Medical Research Council (Grants GNT1113531 and GNT2000001). The research conducted at the Murdoch Children's Research Institute was supported by the Victorian Government's Operational Infrastructure Support Program.

**Competing interests** The authors declare no competing interests. YB and MC are co-Founders of Genetics Adviser, Inc.

**Patient and public involvement** Patients and/or the public were involved in the design, or conduct, or reporting, or dissemination plans of this research. Refer to the Methods section for further details.

**Patient consent for publication** Not applicable.

**Provenance and peer review** Not commissioned; externally peer reviewed.

**ORCID iDs**
Sophie E Bouffler http://orcid.org/0000-0002-4749-4883
Yvonne Bombard http://orcid.org/0000-0002-9516-4539

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
