## [Reviewer comments · BMJ Open]

ARTICLE DETAILS

TITLE (PROVISIONAL)	A study protocol for the two-step offer and return of multiple types of additional genomic findings to families after ultra-rapid trio genomic testing in the acute care setting
AUTHORS	Bouffler, Sophie; Lee, Ling; Lynch, Fiona; Martyn, Melissa; Lynch, Elly; Macciocca, Ivan; Curnow, Lisette; McCorkell, Giulia; Lunke, Sebastian; Chong, Belinda; Marum, Justine E; Delatycki, Martin; Downie, Lilian; Goranitis, I; Vears, Danya F; Best, Stephanie; Clausen, Marc; Bombard, Yvonne; Stark, Zornitza; Gaff, Clara

VERSION 1 – REVIEW

REVIEWER	Michelle Peter Great Ormond Street Hospital for Children NHS Foundation Trust
REVIEW RETURNED	16-Mar-2023

GENERAL COMMENTS	Thank you for giving me the opportunity to review this paper: A study protocol for the two-step offer and return of multiple types of additional genomics findings to families after ultra-rapid trio genomic testing in the acute care setting. This protocol describes an extensive multi-site study that will address stakeholder views on the offer of AF to families via a two-step delivery model. This work is needed as current guidance around offering AF in this way is limited. The findings could, therefore, have important implications for both policy development and subsequent clinical practice. General comments: The Introduction is well-rounded and there is an obvious rationale for the study. The Methods section is impressively comprehensive. The inclusion of Figure 1 is helpful for providing an overview of the study design. I also liked that Table 1 displays the content of each of the surveys across the timepoints, making it easy for the reader to understand what is being measured and at which point. When a Methods section is written as articulately and as detailed as this, it makes replication so much more straightforward so I must commend the authors here. Explanation of the parent and healthcare professional interviews is thorough; the link between the topics to be covered and how this will answer the study questions is clear. Finally, the Discussion provides a considerate analysis of how the results from this study could be used to inform the implementation of a routine service where AF are provided via a two-step delivery model. In sum, this is strong piece of work; I really have only one minor comment concerning the abstract and one typo to point out. Once these changes are made, I believe this protocol paper to be ready
--

	for publication and expect it to be well received by the readership of BMJ Open. Specific comments: Qu2. The abstract is mostly fine, but I think a bit more detail would be helpful here: Will the study focus on parents' motivations for opting in or out of AFs? Or, will it explore parents' experiences of the consent process? Or both? Just as it states that HCPs views on acceptability and feasibility of AF will be examined, a little more clarity around what will be evaluated from the perspective of parents would be great. Page 16; Line 18: Typo – says define, but should say decline.
--	---

REVIEWER	Katharine Press Callahan University of Pennsylvania
REVIEW RETURNED	02-Apr-2023

GENERAL COMMENTS	This is a thoughtful protocol that fills a gap of increasing importance as more neonates have their genomes sequenced. We will need to examine different approaches for evaluating and disclosing AF-- and this is one promising approach. The background thoughtfully sets the stage for this study's approach, making clear the benefits of both AF testing and deferring the offer for AF testing. Specific Comments  -Page 8, line 60: What types of clinical scenarios? Would be helpful to make this more concrete or provide examples. -Page 9, line 20: The exclusion of families for psychological reasons could be problematic, as distressed families may experience the offer of AF or receipt of results differently. To assess the degree and likelihood of this problem, it would be helpful to clarify what "exceptional circumstances" means in this context. -Page 12, line 47: Gene lists- It would be helpful to provide a bit more background about how genes are selected. E.g. how is "treatable" defined? This information may be publicly available elsewhere, but given its centrality to the acceptability of AF testing would be good to include it here. -Page 13, line 40: How was the decision not to report VUS made? Other Comments:  -What supports are offered to parents who may become distressed upon discussing these topics, which seems the greatest risk for parents of children who have died? -Which genes are tested for is an important part of this protocol. I suggest you expand the discussion of how genes are selected and move this earlier—i.e. mention in abstract, set up in background, etc. -Will it be possible to link records between survey time points? -Please provide some justification for the sample size – power calculation, qualitative saturation etc. -How long after being approached about AF will parents participate in the survey or interview? Minor points  - Page 5, Line 25- This study will evaluate not only practicability but also acceptability/value -Page 5, line 57: Missing oxford comma after status -Page 10, line 30: Would restructure sentence as it's unclear whether the tool or the appointment are called "Genetic Adviser." -Page 10, line 32: ADVISER- is this capitalization necessary, and if it
---

	stands for some acronym might make sense to provide the full acronym on first mention
--	---

VERSION 1 – AUTHOR RESPONSE

Reviewer: 1

Dr. Michelle Peter, Great Ormond Street Hospital for Children NHS Foundation Trust

Specific comments:

Qu2. The abstract is mostly fine, but I think a bit more detail would be helpful here: Will the study focus on parents' motivations for opting in or out of AFs? Or, will it explore parents' experiences of the consent process? Or both? Just as it states that HCPs views on acceptability and feasibility of AF will be examined, a little more clarity around what will be evaluated from the perspective of parents would be great.

We have added a line in the abstract to expand on parental evaluation. "Evaluation will focus on parental preferences, uptake, decision support use, and understanding of AF."

Page 16; Line 18: Typo – says define, but should say decline.
Thank you for identifying this typo, it has been corrected.

Reviewer: 2

Dr. Katharine Press Callahan, University of Pennsylvania, The Children's Hospital of Philadelphia Research Institute

Specific Comments

-Page 8, line 60: What types of clinical scenarios? Would be helpful to make this more concrete or provide examples.

Wording has been changed to clarify. "Research has shown that parents commonly try to conceive shortly after experiencing a perinatal loss.^{32,33} To best support families in this situation, parents who enquire seeking recurrence or reproductive risk information before the study-prompted offer of AF may have the opportunity for earlier analysis."

-Page 9, line 20: The exclusion of families for psychological reasons could be problematic, as distressed families may experience the offer of AF or receipt of results differently. To assess the degree and likelihood of this problem, it would be helpful to clarify what "exceptional circumstances" means in this context.

We have added some examples to the text: "Examples of such circumstances may be families experiencing high levels of psychological distress, cases where the parents are in the process of relinquishing care of the child, or if one parent is no longer involved in the child's care."

-Page 12, line 47: Gene lists- It would be helpful to provide a bit more background about how genes are selected. E.g. how is "treatable" defined? This information may be publicly available elsewhere, but given its centrality to the acceptability of AF testing would be good to include it here.

No new gene lists were created for this study. The gene lists used are entirely based on gene lists developed in prior studies, with the references provided describing in detail the process of gene

selection. These typically involved semi-structured frameworks and large groups of experts. We have inserted a sentence at the start of the gene list paragraph to clarify this: “All gene lists have been developed in previous studies, using structured frameworks and expert groups to define content.”

-Page 13, line 40: How was the decision not to report VUS made?

This type of testing is classified as population screening. Guidelines for population screening generally discuss reporting pathogenic variants only as the individual being tested is unaffected/asymptomatic. A VUS cannot be used to inform any clinical care. This is consistent with the recommendations by the ACMG (Murray et al. 2021), which we have now clarified in the text: “Variants of uncertain significance will not be reported, in line with population screening guidelines.⁴⁶”

Other Comments:

-What supports are offered to parents who may become distressed upon discussing these topics, which seems the greatest risk for parents of children who have died?

All families will have had prior contact with clinical genetics services. Before contacting any families, the study team will confirm with the referring clinicians whether the child is deceased and/or if there are other reasons to postpone contact. The study team have designed two pathways in Genetics Adviser, one specifically for families whose child is deceased and excludes the offer of analysis for childhood-onset conditions. This is stated in the Decision Support section, and we have modified the text to make this more clear: “To minimise distress to parents of children who have died, an alternative version of the tool will be used for these parents, which omits the section on paediatric-onset additional findings.”

Throughout participation in the study, parents are given access to genetic counsellors. At the top of all surveys, parents are provided with information on who to contact if the contents cause any distress. Genetic counsellors are specifically trained to handle situations such as these, and where possible families will see counsellors they already have a relationship with from the initial diagnostic testing.

-Which genes are tested for is an important part of this protocol. I suggest you expand the discussion of how genes are selected and move this earlier—i.e. mention in abstract, set up in background, etc.

Please refer to our previous response. As we did not develop any new gene lists this is not a prominent part of the study. All gene lists are publicly available and based on previously published work which has been referenced where appropriate.

-Will it be possible to link records between survey time points?

Yes, all longitudinal data will be matched to the original record for the child and hence can be paired for evaluation. We have noted this in the methods: “Survey responses will be linked between timepoints.”

-Please provide some justification for the sample size – power calculation, qualitative saturation etc.

The offer of Additional Findings is being made to a cohort recruited separately for ultra-rapid diagnostic testing. As such, the sample size has been determined by recruitment for the broader study. We acknowledge this may be a limitation and larger purposefully designed studies may be required in the future. This will be a consideration when publishing findings from the study, and we have noted this in the discussion: “The sample size for this protocol is pre-determined by recruitment into the larger Acute Care Genomics study. This will be considered when reporting the findings of our

study and we acknowledge that there may still be a need for larger, purposefully designed studies in the future.”

-How long after being approached about AF will parents participate in the survey or interview?

Please refer to Figure 1 to see the study flow including survey and interview timepoints. Surveys will be sent throughout the study. The approach for interviews will be made a minimum of three months after the results have been returned. We have added this information in the text: “Those who do so will be contacted a minimum of three months after result return or after they decline AF and offered the option to participate in a semi-structured interview, either by phone or via an online platform (see Figure 1).”

Minor points

- Page 5, Line 25- This study will evaluate not only practicability but also acceptability/value

We have amended the manuscript as follows: “This study will investigate the practicability, acceptability, and value of offering AF analysis to parents and children using stored genomic data 3-6 months after diagnostic testing.”

-Page 5, line 57: Missing oxford comma after status

Corrected

-Page 10, line 30: Would restructure sentence as it's unclear whether the tool or the appointment are called “Genetic Adviser.”

We have restructured the sentence. The sentence now reads; “The study will deploy an interactive web-based decision support tool, called Genetics Adviser,³⁴ as an adjunct to a genetic counselling appointment.”

-Page 10, line 32: ADvISER- is this capitalization necessary, and if it stands for some acronym might make sense to provide the full acronym on first mention

The capitalisation does form an acronym. We have now added this in the manuscript: “The tool is an expanded version of the Genomics ADvISER, (a genomics decision AiD about Incidental SEquencing Results)...”